# Detection of Wildfire Smoke Images Based on a Densely Dilated Convolutional Network

**Tingting Li** [1,2] , **Enting Zhao** [1,2], **Junguo Zhang** [1,2,*] **and Chunhe Hu** [1,2,*]

[1]  School of Technology, Beijing Forestry University, Beijing 100083, China; litingting@bjfu.edu.cn (T.L.); zhao_enting@163.com (E.Z.)
[2]  Key Lab of State Forestry and Grassland Administration for Forestry Equipment and Automation, Beijing 100083, China
*  Correspondence: zhangjunguo@bjfu.edu.cn (J.Z.); huchunhe@bjfu.edu.cn (C.H.)

**Abstract:** Recently, many researchers have attempted to use convolutional neural networks (CNNs) for wildfire smoke detection. However, the application of CNNs in wildfire smoke detection still faces several issues, e.g., the high false-alarm rate of detection and the imbalance of training data. To address these issues, we propose a novel framework integrating conventional methods into CNN for wildfire smoke detection, which consisted of a candidate smoke region segmentation strategy and an advanced network architecture, namely wildfire smoke dilated DenseNet (WSDD-Net). Candidate smoke region segmentation removed the complex backgrounds of the wildfire smoke images. The proposed WSDD-Net achieved multi-scale feature extraction by combining dilated convolutions with dense block. In order to solve the problem of the dataset imbalance, an improved cross entropy loss function, namely balanced cross entropy (BCE), was used instead of the original cross entropy loss function in the training process. The proposed WSDD-Net was evaluated according to two smoke datasets, i.e., WS and Yuan, and achieved a high AR (99.20%) and a low FAR (0.24%). The experimental results demonstrated that the proposed framework had better detection capabilities under different negative sample interferences.

**Keywords:** wildfire smoke detection; CNN; DenseNet; dense block; candidate smoke region; dilated convolution; cross entropy loss

## 1. Introduction

Wildfires not only destroy the natural ecological environment, but also threaten human safety and property [1]. Since image-based fire detection effectively reduces outside interference compared to the currently available sensors, image-based fire detection has become a hot topic in modern wildfire alarm systems [2]. Fire is often accompanied by smoke, which is emitted faster than flames. Therefore, smoke detection is an effective way to recognize potential fire disasters at the beginning of a breakout.

Existing methods regarding automatic smoke detection can be itemized into two categories, namely conventional detection approaches based on shallow machine learning [3–5], and deep learning methods based on deep neural networks [6]. Conventional detection approaches generally use handcrafted smoke features to train the classifiers (e.g., K-nearest-neighbor (KNN), support vector machine (SVM), AdaBoost) in the training and test scenes. Deep learning-based methods have been widely applied to visual detection tasks, such as wildlife identification [7], disease detection [8], pedestrian detection [9], etc. Extensive studies on smoke detection using convolutional neural networks (CNNs) have been shown to be capable of learning representative and essential features of smoke images and have presented ideal detection accuracy [10]. Due to the complex backgrounds of wildfire smoke images, the accuracy of many existing image-based smoke detection techniques is still

inconsistent. In previous studies that used tailored smoke images for training, there would have been errors when testing the entire image.

Recent progress regarding smoke image detection, such as background processing [11], demonstrated that CNNs could deal with complex background interference and achieve high accuracy. However, it is still infeasible to apply these technologies directly to wildfire smoke detection. In order to analyze the interference of smoke-colored regions like cloud, fog, and haze on wildfire smoke detection, images with pedestrians and vehicles are not used as non-smoke samples. Smoke-colored regions resulting from fog, cloud, or haze are similar to smoke in color, texture, and shape features. Therefore, it is difficult to discriminate smoke from smoke-colored regions, thereby causing a high false-alarm rate. Another crucial issue is the lack of wildfire smoke images. There are not many wildfires and so wildfire smoke images cannot be collected often. The lack of wildfire smoke images leads to a serious training dataset imbalance, which results in over-fitting.

To address the above challenges, we propose a novel framework integrating conventional methods into CNN for wildfire smoke detection, consisting of a candidate smoke region segmentation strategy and advanced network architecture, namely wildfire smoke dilated DenseNet (WSDD-Net). In order to ensure that color information was not affected by illumination changes, RGB images were first converted into the YUV color space. Then, candidate smoke regions were segmented from wildfire smoke images using a segmentation strategy, which is defined in Section 3, and the segmented images were directly trained in WSDD-Net. A balanced cross entropy (BCE) loss function was also employed during training according to the proportion of smoke images in the training data, in order to weaken the imbalanced classes problem.

The rest of the paper is organized as follows. Section 2 gives a short summary of fire detection and smoke detection. The detailed segmentation strategy and WSDD-Net architecture is described in Section 3. Section 4 presents our evaluation methods and results compared with some other CNN architectures. Section 5 presents the conclusions of this paper.

## 2. Related Work

Fire detection-related studies have aroused much more attention in recent years. Yang et al. [12] established a model based on the Levenberg–Marquardt back propagation neutral network to recognize fire status. In Zhang, Shen, and Zou [13], a video-based fire recognition probability method based on color and motion features was presented. Smoke detection is another method of fire detection, which aims to improve the detection rate and reliability. Tung et al. [14] employed a median method combined with a fuzzy c-means method to segment moving regions and cluster candidate smoke regions from moving regions. Jia et al. [15] proposed a method of segmenting smoke regions based on saliency detection. Yuan [16] used a histogram sequence of local binary pattern (LBP) and local binary pattern variance (LBPV) pyramids to detect smoke.

All of the above detection algorithms mainly used handcrafted features and employed shallow machine-learning methods. A cost-effective smoke detection framework based on deep learning was proposed by Muhammad et al. [17] and focused on detection accuracy in complex backgrounds. In Mao, Wang, Dou, and Li [18], a smoke detection method based on multi-channel CNNs with an accuracy of 98% was introduced. Sharma et al. [19] used two pre-trained CNNs, namely VGG-16 and ResNet-50, to train an imbalanced smoke image dataset for fire detection. In order to avoid over-fitting due to lack of smoke images and imbalanced training data, Xu et al. [20] used synthetic smoke images to expand the smoke dataset and domain adaption to train the CNN structures. Zhang et al. [21] used synthetic wildfire smoke images to train an end-to-end object detection framework and a faster RCNN, which demonstrated the feasibility of synthetic smoke images to expand the dataset. However, this method of expanding datasets with synthetic wildfire smoke images consumes manpower, and it takes an excessive amount of time for the data size to meet the training requirements.

Recent works have shown that if CNNs contain connections between the previous output and input layers, the CNNs can be deeper, more accurate, and more efficient for training. Huang et al. [22]

solved this problem and proposed the DenseNet, which connected each layer to every other layer in a feed-forward manner. Compared with classical deep learning methods, DenseNet not only performs better regarding wildfire smoke detection, but also achieves better results in false-alarm rate and accuracy. It alleviates the problem of the vanishing-gradient, enhances feature propagation, and reduces the number of parameters and training time by feature reuse.

Almost all of the models used the cross entropy loss function, but it is easy to over-fit when there is a dataset class imbalance. The focal loss function was proposed to solve this issue [23]. Abulnaga and Rubin [24] developed fully convolutional neural network models for the segmentation of ischemic stroke lesions, and made use of the focal loss function, which demonstrated the ability to identify finer features by focusing on hard-to-classify examples.

Inspired by recent use of focal loss in image-based object detection, we improved the cross entropy loss function to be suitable for wildfire smoke detection. We also added negative samples, such as cloud, fog, and haze, as interference to our dataset. Unlike the aforementioned works, our network directly trained candidate smoke regions to eliminate the interference of complex backgrounds in the feature extraction process. In addition, we attempted to address the problems of high false-alarm rates and imbalanced training sets simultaneously.

## 3. Methods

Our goal was to reduce the false-alarm rate in detecting wildfire smoke, alleviate the imbalance of the wildfire smoke dataset, and undertake a novel framework integrating conventional methods into CNN for wildfire smoke detection. The framework was divided into two major phases. First, a segmentation strategy of candidate smoke regions was introduced as the pre-processing method of separating smoke and non-smoke images. Second, the candidate smoke regions were detected by our proposed WSDD-Net.

### 3.1. Candidate Smoke Region Segmentation

Smoke images show different results in each color space due to the change in illumination and environment [25]. Past research discussed smoke images in different color spaces [26–28]. The results showed that color information is least affected by illumination changes in the YUV color space. According to the probability statistics of thousands of real smoke images, Prema, Vinsley, and Suresh [25] demonstrated that the U-V of non-smoke regions was mainly distributed from 0 to 40, while candidate smoke regions were mainly distributed between 40 and 130. The segmentation strategy is shown in the following formula.

$$S_1(x,y) = \begin{cases} 1 & if \left| U(x,y) - 128 \right| > T_U \\ 0 & Otherwise \end{cases}, \tag{1}$$

$$S_2(x,y) = \begin{cases} 1 & if \left| U(x,y) - V(x,y) \right| > T_{UV} \\ 0 & Otherwise \end{cases} \tag{2}$$

where $U(x,y)$ and $V(x,y)$ are images of the U and V components, $T_U$ and $T_{UV}$ are the threshold values of Equation (1) and Equation (2), which were derived from several experiments, and $S_1(x,y)$ and $S_2(x,y)$ are the pixels' satisfying rules. Finally, candidate smoke regions were determined by using Equation (3).

$$S_0(x,y) = \begin{cases} I_i(x,y) & if S_1(x,y) = 1 (or) S_2(x,y) = 1 \\ 0 & Otherwise \end{cases}, \tag{3}$$

where $I_i(x,y)$ is the input RGB image and $S_0(x,y)$ is the segmented candidate smoke region.

Figure 1 shows the segmentation results in the YUV color space. It was obvious that the segmentation strategy removed the complex background, except for smoke-colored regions including

elements like cloud, fog, or haze. For the sake of eliminating the interference of a non-smoke region such as sky, cloud, fog, and haze, further classification of the candidate smoke region is needed.

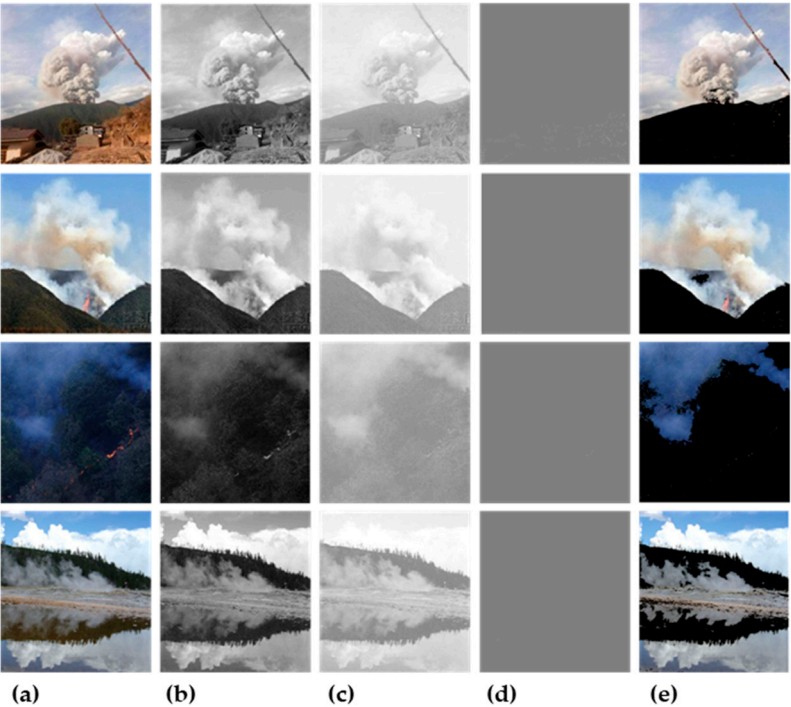

**Figure 1.** Segmentation results in the YUV color space. (**a**) RGB image, (**b**) Y component, (**c**) U component, (**d**) V component, and (**e**) candidate smoke region.

### 3.2. Proposed WSDD-Net

In order to further classify the candidate smoke regions, we proposed a novel deep learning architecture, i.e., dilated DenseNet, for wildfire smoke detection. Below, we introduce the major structure of DenseNet, followed by a detailed description of our proposed wildfire smoke dilated DenseNet (WSDD-Net) architecture. The WSDD-Net architecture is presented in Figure 2. The dilated dense block in Figure 2 replaced the common convolutions in the original DenseNet, resulting in dilated convolutions.

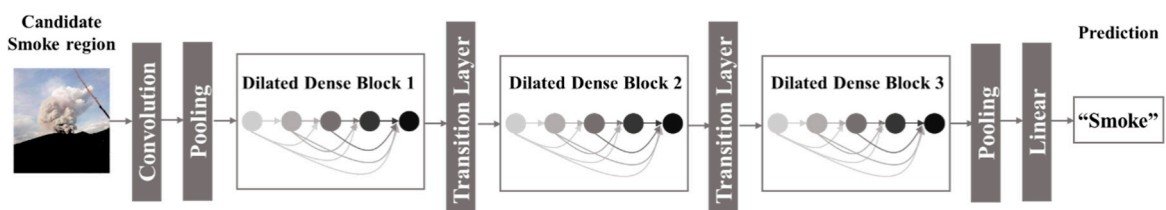

**Figure 2.** Network architecture of proposed wildfire smoke dilated DenseNet (WSDD-Net).

#### 3.2.1. DenseNet

The WSDD-Net was inspired by the state-of-the-art network architecture, i.e., DenseNet [22]. DenseNet is made up of many dense blocks. Each layer takes all previous outputs as input, each dense block is connected to a transition layer, and the last layer is connected to the global pooling and the fully connected layer directly. Figure 3 illustrates the architecture of the original dense block. Batch normalization (BN), activation function, pooling, and convolution are defined as the composite

function $F_l(\cdot)$. We denote the output of the $l^{th}$ layer as $x_l$. The $l^{th}$ layer receives the outputs of all previous layers as input

$$x_l = H_l([x_0, x_1, \dots, x_{l-1}]),\tag{4}$$

where $x_0, x_1, \dots, x_{l-1}$ denote the outputs of $0, 1, \dots, l-1$ layers.

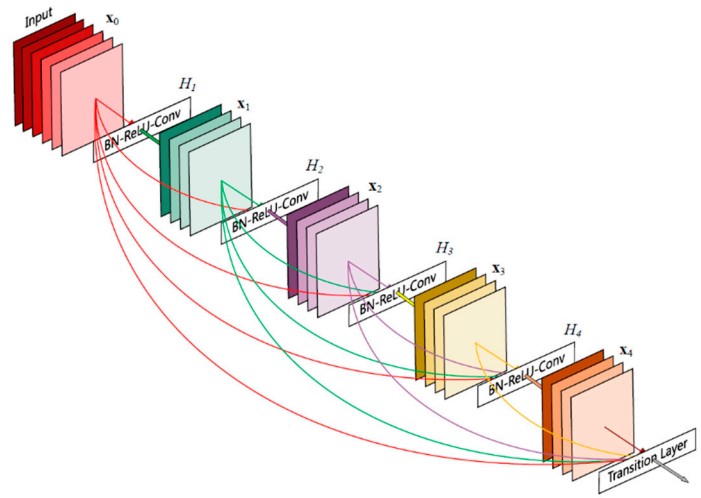

**Figure 3.** A five-layer dense block.

### 3.2.2. Dilated Dense Connection (DDC)

Using the dilated convolutions, multi-scale features were extracted without extra computational cost compared to that of approaches using common convolution kernels [29]. Taking the advantages into account, the DDC replaced the common convolutions in the original DenseNet, resulting in dilated convolutions. The original dense block achieved multi-scale feature extraction by stacking $3 \times 3$ convolutions. As the dilated convolutional had a larger receptive field compared to the common convolution [30], the proposed DDC block adopted the dilated convolutions to achieve multi-scale feature extraction. As shown in Figure 4, dilated convolutions with two dilation rates, i.e., 2 and 3, were involved in the proposed DDC block. The common $3 \times 3$ convolution was placed after each dilated convolution to fuse the previous outputs.

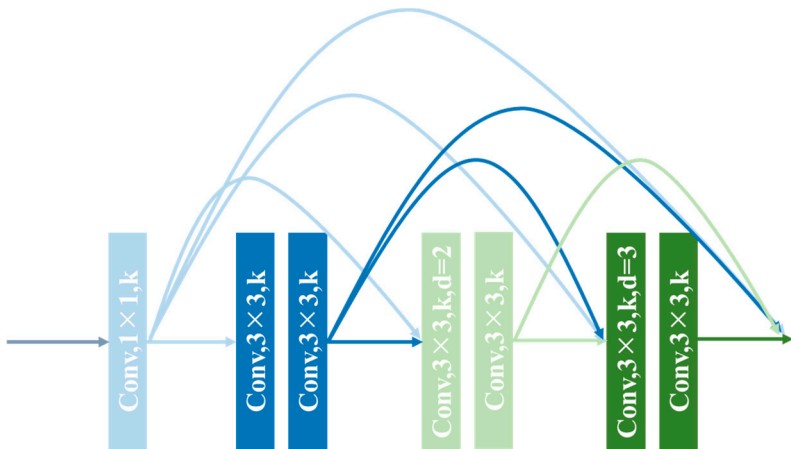

**Figure 4.** Framework of the proposed dilated dense connection (DDC).

The proposed DDC resolved two crucial shortcomings of the existing structure. First, compared to that of the proposed DDC block, the dilation rates of existing structures are usually 4, 8, 16, etc. Because

of these large numbers, the receptive field of existing structures normally exceeded the image size. As a result, convolution computation required a large number padding zeros. Second, the architecture of the prior framework had no short-cut connections, which means it was unable to perform multi-scale feature extraction.

### 3.3. Loss Function

We trained the network using an improved cross entropy loss function, namely balanced cross entropy (BCE), which was inspired by the focal loss function [23]. It was designed for training to account for the imbalance in wildfire smoke datasets by down-weighting the non-smoke part of the dataset. The improved loss function was introduced from the cross entropy loss for binary classification:

$$CE(p, y) = \begin{cases} -\log p & y = 1 \\ -\log(1-p) & y = 0 \end{cases}, \tag{5}$$

where $y = \{0, 1\}$ is the real image label and $p \in [0, 1]$ is the predicted class probability for the class with the label $y = 1$. In order to address class imbalance, a weighting factor $\alpha \in [0, 1]$ for class 1 and $1 - \alpha$ for class 0 was introduced. $\alpha$ was treated as a hyper-parameter to set cross validation. We wrote the balanced cross entropy loss as

$$BCE(p, y) = \begin{cases} -\alpha \log p & y = 1 \\ -(1-\alpha) \log(1-p) & y = 0 \end{cases}. \tag{6}$$

This loss function alleviated the problem of the wildfire smoke dataset imbalance while guaranteeing a high detection accuracy rate and low false-alarm rate.

## 4. Experimental Results

The proposed WSDD-Net was established using the Pytorch toolbox. The experiment system environment was Ubuntu 16.04 and the programming language was python. The hardware configuration consisted of an E5-2620 CPU (Intel Corporation, Santa Clara, CA, USA) and a GeForce GTX 1080ti GPU (Nvidia Corporation, Santa Clara, CA, USA). The network was trained with a mini-batch size of 32. In order to achieve faster network convergence, the initial learning rate was set to 0.1. After that, the learning rate decreased by 0.1 times per 20 epochs. The network input materials were RGB color images, which were resized to $224 \times 224$.

### 4.1. Wildfire Smoke Dataset

The wildfire smoke (WS) dataset consisted of 4595 images (1685 wildfire smoke images and 2910 cloud, fog, and haze images), which was collected from the literature and websites by the authors. From the smoke dataset and the non-smoke dataset, 80% of images were selected, respectively. These smoke images and non-smoke images were randomly separated according to the ratio of 3:1 for training and validation. The remaining 20% of the images were prepared as the testing set for the performance comparison between the benchmark algorithms and our framework. Rotation and mirroring were used to expand the size of the training dataset. Smoke and non-smoke images in the training dataset were rotated by 180° and then reflected vertically to generate additional images, resulting in an augmented training dataset with 11,028 images. Examples of different categories of the WS dataset are shown in Figure 5. The first two rows show the segmented patches of wildfire smoke image patches. The third and fourth rows show fog and cloud images patches, respectively. It can be seen that illumination, shooting distance, smoke color and background were different due to the different sources of these images.

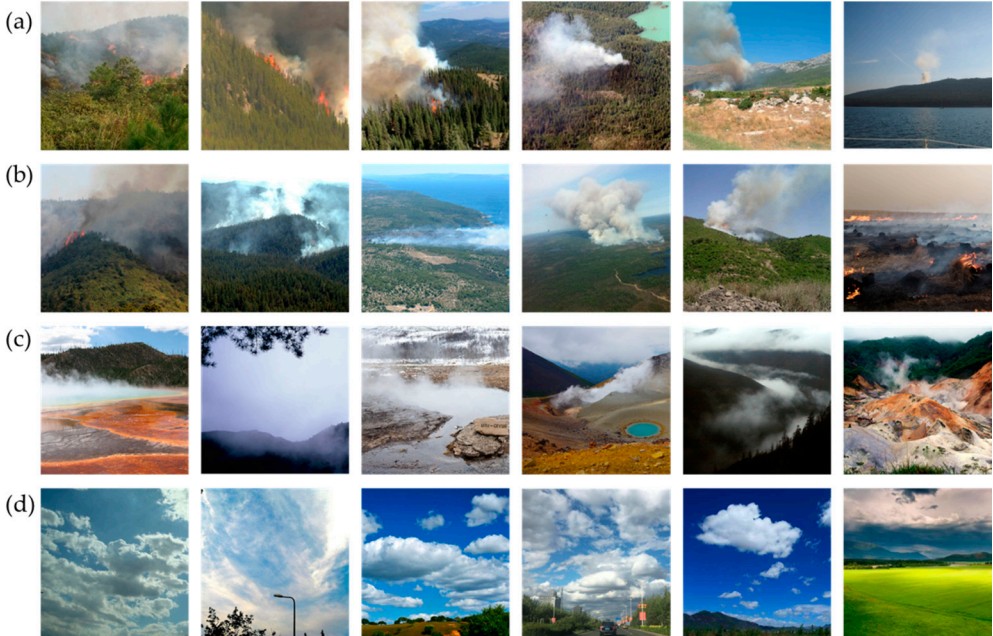

**Figure 5.** Sample images from the training and testing sets. (**a**) Smoke image, (**b**) smoke image, (**c**) fog image, and (**d**) cloud image.

### 4.2. Evaluation Criterion

To evaluate the performance of the proposed method, we uses a train–validation–test scheme. The training set was used for the actual training of the method, while fine-tuning the hyper-parameters was carried out using the validation set. The total performance of proposed framework was evaluated using the wildfire smoke testing set. When evaluating models for binary classification on a given dataset of positives and negatives, four different types of data are usually defined: true positive (TP), true negative (TN), false positive (FP), and false negative (FN). TP is the number of true positives, i.e., the number of smoke patches classified as smoke, TN is the number of true negatives, i.e., the number of non-smoke patches classified as non-smoke, FN is the number of false negatives, i.e., the number of non-smoke patches classified as smoke, and FP is the number of false positives, i.e., the number of smoke patches classified as non-smoke. The accuracy rate (AR), false-alarm rate (FAR), and recall rate (RR, defined below) were adopted as the criteria for the performance evaluation. In some extreme cases, RR and AR were conflicting, which required the use of an F1-score (F1) to evaluate the performance of the algorithm [18]. The definitions are as follows.

$$AR = \frac{TP + TN}{TP + TN + FP + FN}, \tag{7}$$

$$FAR = \frac{FN}{TN + FN}, \tag{8}$$

$$RR = \frac{TP}{TP + FN}, \tag{9}$$

$$F1 = \frac{2(RR \times DR)}{RR + DR}, \tag{10}$$

where

$$DR = \frac{TP}{TP + FP}. \tag{11}$$

### 4.3. Results of the WS Dataset

To demonstrate the advantages produced by proposed WSDD-Net, several well-known deep learning networks like ResNet-50/101 [31], Incveption-V3 [32], and DenseNet-121/169 [22], were also used for the performance evaluation.

The training processes of these networks are illustrated in Figure 6. Figure 6a shows the changing accuracy of the system with training over different epochs. It was clear that the AR of the proposed WSDD-Net rapidly rose from 0 to 5 epochs and gradually rose from 5 to 10 epochs. Then, the AR reached 99% and fluctuated around 99%. Figure 6b shows the changing loss of the training set. It was obvious that the loss of WSDD-Net sharply dropped from 0 to 5 epochs and slightly descended from 5 to 10 epochs. Following this, the loss varied gradually and fluctuated between 0.05 and 0.09. Although the WSDD-Net had a better performance than other classical networks in training, the critical performance of the network for wildfire smoke detection depended on the results of the testing phase.

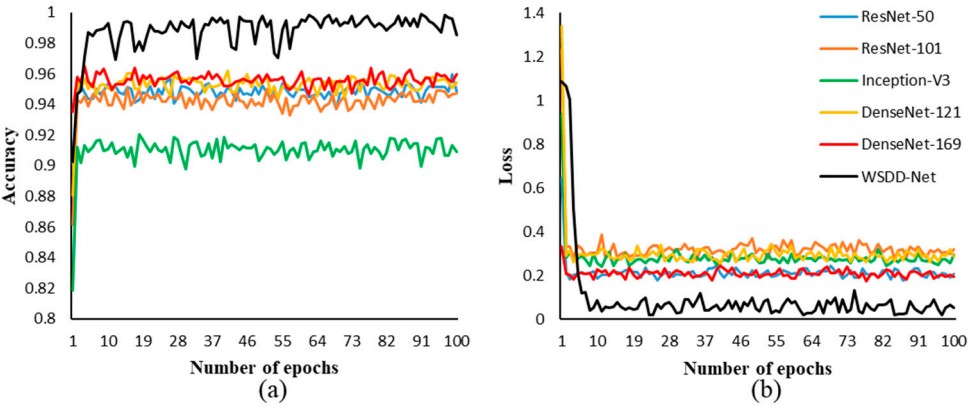

**Figure 6.** Training process curves of ResNet-50/101, Inception-v3, DenseNet-121/169, and WSDD-Net. (**a**) Accuracy curve and (**b**) loss curve.

The results for the wildfire smoke test set are listed in Table 1. It was clear that the proposed WSDD-Net achieved the highest AR (99.20%) and F1 (99.25) among the models trained on our dataset, which was 1.91% and 1.78 higher than the runner-up, DenseNet-169. Furthermore, we noticed that almost all of the listed classical algorithms performed poorly when dealing with the images of non-smoke dataset, i.e., the lowest FAR produced by the ResNet-101 was 2.41%, which was seven times higher than the FAR of WSDD-Net. The test time of our WSDD-Net was 9.93 ms, which was competitive compared to that of the best performance. The DenseNet-121 and WSDD-Net have much smaller model sizes than other networks, and have the ability to adopt small datasets to prevent overfitting. Compared to the DenseNet-121, the proposed WSDD-Net uses multiple dilated convolutions to extract multi-scale features. As shown in Table 1, the proposed WSDD-Net outperformed the DenseNet-121 and produced the best F1. The experimental results showed that the proposed novel framework integrating conventional method into CNN maintained a good balance between network size and feature learning capacity, which is extremely effective for small-scale wildfire smoke datasets.

**Table 1.** Comparisons with classical deep convolutional neural networks (CNNs).

| Model | Size (MB) | AR (%) | FAR (%) | F1 | RR (%) | DR (%) | Test time (ms) |
|---|---|---|---|---|---|---|---|
| ResNet-50[30] | 89.69 | 96.02 | 2.41 | 96.23 | 97.85 | 94.66 | 10.31 |
| ResNet-101[30] | 162.14 | 95.54 | 7.56 | 95.94 | 93.77 | 98.22 | 12.89 |
| Inception-V3[31] | 83.12 | 91.72 | 15.12 | 92.68 | 88.20 | 97.63 | 10.70 |
| DenseNet-121[22] | 26.53 | 96.34 | 4.47 | 96.60 | 96.18 | 97.03 | 8.17 |
| DenseNet-169[22] | 47.64 | 97.29 | 2.75 | 97.47 | 97.62 | 97.33 | 10.07 |
| WSDD-Net | 39.72 | 99.20 | 0.34 | 99.25 | 99.70 | 98.81 | 9.93 |

To verify the availability of our proposed WSDD-Net, statistical measures were also used to evaluate model performance, which was inspired by the paired *t*-test used in Ghaffari et al. [33]. Detailed results of the experiments are shown in Figure 7 and Table 2. It was observed that the proposed method exhibited excellent performance in both AR and FAR, showing large advantages compared to other algorithms. These results indicated the stability of our model. Table 2 represents the statistical significance from comparing the average AR of each model using an independent *t*-test. A significance of <0.05 was considered to be statistically significant. As shown in Table 2, the greater the t-value, the more significant the difference between the two models. The results demonstrated that WSDD-Net had a higher average AR and was more suitable for wildfire smoke detection than other classical networks from a statistical perspective.

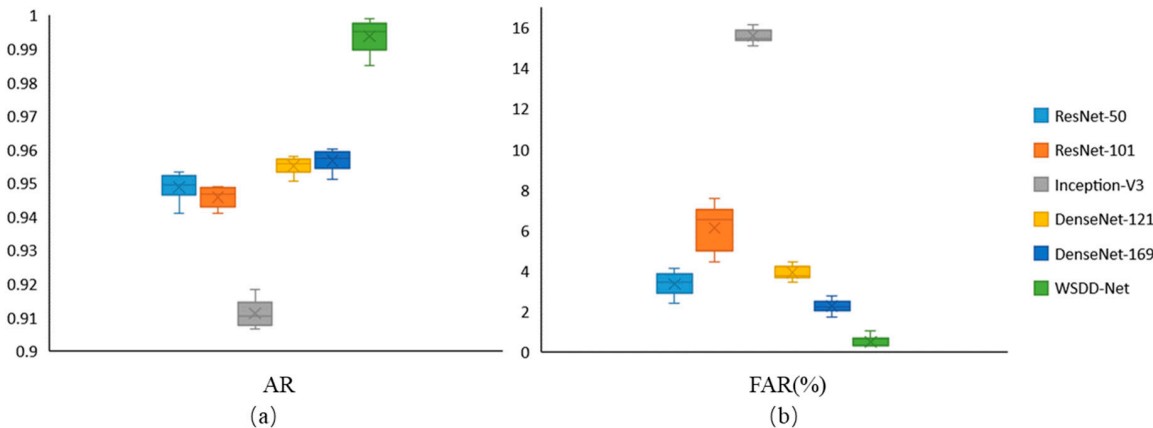

**Figure 7.** Test results from the wildfire smoke (WS) dataset. (**a**) Box-plot for 10 AR values of each method and (**b**) box-plot for 10 FAR values of each method.

**Table 2.** Statistical significance of performance comparisons.

| Null Hypothesis($H_0$) | T-value | Sig |
|---|---|---|
| ResNet-50[30] = ResNet-101[30] | 1.988 | 0.063 |
| DenseNet-121[22] = DenseNet-169[22] | 1.349 | 0.194 |
| WSDD-Net = Inception-V3[31] | 41.504 | 0.000 |
| WSDD-Net = DenseNet-121[22] | 22.338 | 0.000 |
| WSDD-Net = DenseNet-169[22] | 20.863 | 0.000 |
| WSDD-Net = ResNet-50[30] | 23.186 | 0.000 |
| WSDD-Net = ResNet-101[30] | 27.116 | 0.000 |

### 4.4. Results on the Yuan Dataset

To further illustrate the excellent performance of the proposed framework, we evaluated the framework using a public smoke dataset called the Yuan dataset [34]. The algorithms for the comparisons included MCCNN [1], DNCNN [34], ZF-Net [35], and HLTPMC [36].

The experimental results are presented in Figure 8. The proposed WSDD-Net achieved the highest AR (99.71%) and RR (99.64%) among the five networks using the Yuan dataset, and obtained a higher F1 than the other methods, except for MCCNN. Although our method obtained a lower F1 than MCCNN, the FAR of our network was the lowest. In conclusion, the DDC structure produced significant improvements in both AR and FAR for our WSDD-Net. Better experimental results were obtained using the WS and Yuan datasets, indicating that the framework integrating conventional methods into CNN for wildfire smoke detection had better smoke detection capabilities under different negative sample interferences and achieved a high accuracy rate and a low false-alarm rate.

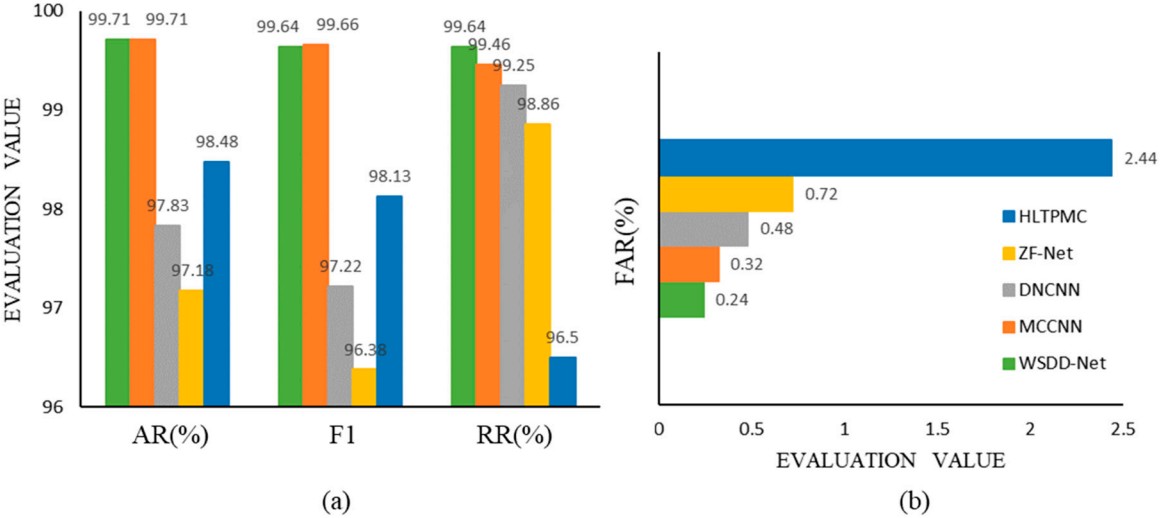

**Figure 8.** Experiment results using the Yuan dataset.

## 5. Conclusions

Many researchers have attempted to use CNN for wildfire smoke detection. However, the application of CNNs in wildfire smoke detection still faces several issues. For example, almost all smoke datasets have a serious class imbalance. It is difficult for deep neural networks to directly detect wildfire smoke images due to the interference of candidate smoke regions. To address these issues, we proposed a novel framework integrating conventional methods into CNN for wildfire smoke detection. The framework consisted of a candidate smoke region segmentation strategy and advanced network architecture, namely wildfire smoke dilated DenseNet (WSDD-Net). The candidate smoke region segmentation removed the complex backgrounds of wildfire smoke images. The proposed WSDD-Net achieved multi-scale feature extraction by combining dilated convolutions with dense block. In order to solve the problem of the dataset imbalance, an improved cross entropy loss function, namely balanced cross entropy (BCE), was used instead of the original cross entropy loss function in the training process. The proposed WSDD-Net was evaluated using the WS and Yuan smoke datasets, and achieved a high AR (99.20%) and a low FAR (0.24%). The experimental results demonstrated that the proposed framework had better detection capabilities under different negative sample interferences.

**Author Contributions:** Data curation, E.Z.; project administration, J.Z.; writing—original draft, T.L. and C.H.; writing—review and editing, T.L.

**Funding:** This study was financially supported by the Fundamental Research Funds for the Central Universities (Grant No.2016ZCQ08 and Grant No.2019SHFWLC01).

**Conflicts of Interest:** The authors declare no conflict of interest.

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
