# Peer review of "Detection of Wildfire Smoke Images Based on a Densely Dilated Convolutional Network"

_electronics, doi:10.3390/electronics8101131_

Round 1

Reviewer 1 Report

The authors have presented a novel way to look at the detection of smoke in images that helps to deal with an imbalance in the availability of smoke images and the availability of non-smoke images.

There are several major issues with the manuscript that I will deal with before moving on to discuss more minor points.

The first issue is that the authors haven’t consistently provided enough explanation of what they did in some of the key areas.  For example, on line 170, they mention alpha but do not define it. Is alpha the ratio of smoke images/total images (where total images = smoke + not smoke)? If alpha is less than 1 here, then you have a low penalty for false positives and false negatives.  If there is a lower penalty for guessing that the image contains "smoke" when it does not, then wouldn't this bias your classification accuracy?  If alpha is used in the cost function modification to down-weight the non-smoke class and it classifies it as smoke, there is a decreased penalty for that error.   Am I understanding this correctly? Also on line 176 the authors mention gamma but again, do not define what gamma is.

Some of the issues above may be the result of the translation of the manuscript which may have led to the statements in English not reflecting what the authors actually meant.  However, on the whole, it is difficult overall to understand what they did.  Given that other readers may also be less knowledgeable about deep learning, it would be worthwhile for the authors to explain some of the features of their system in more detail, particularly those items listed above.  

On line 183, you mention that 80% of the images were used in training. Was that 80% of each of the smoke and non-smoke images or 80% of the whole database?  I assume it was 80% of each since taking 80% of the whole database would result in a higher percentage of true negatives for the training, which could lower the FAR.  Similarly, on lines 185-186, you mentioned increasing the sample size by mirroring, rotating and cropping the images.  Here again, the authors have not indicated if this was done to all of the images or just the smoke images.

There are at least two places in the results where the authors have drawn conclusions that are not supported by their statements.  Again, this may only require clarification in 1 or 2 cases.  The first is on line 211 where the authors indicate that their network, WSDD-Net is “more suitable for wildfire smoke detection than other classical networks”.  However the authors are discussing the results of the training.  While the WSDD-Net have better performance in training, the critical performance of the system is with respect to the testing phase. Further on in the sentence on line 212, the authors state that the WSDD-Net was “more suitable … from a statistical perspective”.  The authors did not report any statistical analyses and so they cannot make this claim.  The only data they have provided was the mean performance data for each of the performance metrics.  They could, at least present standard deviations or standard error measures.  Another area where the authors may have drawn a wrong conclusion is for lines 240-243 where they state that the “…framework integrating conventional method to CNN for the wildfire smoke detection has better generalization capabilities…”.  The authors have not indicated that they trained the other networks on the WS database and tested it on the Yuan dataset so that a direct comparison of generalization is possible. 

There are also some general questions about practical issues with using this system to detect fires.  The first is the images in the data set.  Were they images from small or large fires?  The issue here is that the sooner the fire can be detected, the easier it is to extinguish and that would normally imply a small smoke source.  The other issue is even more general.  Have the authors considered any ways in which the location of the fire/smoke could be determined?

The more minor issues with the manuscript are detailed below.

Line 3- “Network” is misspelled in the title and I think that the phrase should be “Densely Dilated” rather than “Dilated Densely” Line 10: replace the word “researches” with “researchers”. Lines 11-15: The sentence “To address 12 these issues, we propose a novel framework integrating conventional method to CNN for the wildfire smoke detection, which consists of a candidate smoke region segmentation strategy, and an advanced network architecture, namely Wildfire Smoke Dilated DenseNet (WSDD-Net).” Should be changed to “To address these issues, we proposed a novel framework for integrating conventional methods to CNN for wildfire smoke detection, which consisted of a candidate smoke region segmentation strategy, and an advanced network architecture, namely Wildfire Smoke Dilated DenseNet (WSDD-Net). Line 28: “Since the image-based fire detection can effectively reduce outside interference compared to the currently available sensors, the image-based fire detection…” should be changed to “Since image-based fire detection can effectively reduce outside interference compared to the currently available sensors, image-based fire detection…”. Line 33: change “Exiting” to “Existing”. Line 37: “…in the train and test scenes. Deep learning based method have been widely applied on…” should be changed to “…in the training and test scenes. Deep learning based methods have been widely applied on…” Line 42: delete the phrase “…to be affected.” from the end of the sentence. Line 43: replace “… used tailored smoke images training directly, thus there would be errors when testing the entire” with “… used tailored smoke images for training, thus there would be errors when testing the entire…”. Line 43: Do the authors mean that the tailored image was only a partial image (that is, smoke only, no background)? Please clarify. Also, did these studies fail to use separate training and testing images? Line 48: Do the authors mean that images with pedestrians or vehicles in them are rejected because those features are inconsistent with a wildfire/forest fire? The other issue here is that in urban areas where you might have people or vehicles in the images, there may well be sources of smoke that don't represent forest fires (such as smoke from industrial operations, smoke from barbecues or even house fires). While there are definitely sources of smoke that may come from within a populated area, they may have nothing to do with a forest fire. Lines 48-49: There should be two separate sentences here. The first sentence should be about the rejection of images with people and/or vehicles in them and the second should be about the difficulties in discriminating smoke from cloud or fog etc. Line 51: I’m not sure if the authors can access the database but there is a database in the United States on the Website “Research Webpage about Smoke Detection for Fire Alarm: Datasets” that contains a section on wildfire smoke images with 12620 images (http://smoke.ustc.edu.cn/datasets.htm). Lines 51-52: rewrite the sentences “On one hand, there are few released wildfire smokedatasets. On the other hand, few wildfire smoke images can be collected.” as “There are few wildfire smoke datasets because there are not many few wildfires and so smoke images cannot be collected.” Line 57: replace “…color information does not affected by illumination…” with “…color information is not affected by illumination…” Line 59: At this point, you should state that this is going to be defined in a following section, otherwise the reader is left wondering what your process was for your segmentation strategy. Line 61-62: “…in order to weaken imbalanced classes problem.” Change to “…in order to weaken the imbalanced classes problem.” Line 63: “The rest of paper is organized as follows.” Should read as “The rest of the paper is organized as follows.” Line 66: Rewrite the sentence “In the end, Section 5 draws conclusions of this paper.” As “Section 5 presents the conclusions of this paper.” Line 68-83: The authors have used an odd way of citing the research that they are discussing in this section and in other places. They begin the sentence with the number referencing the study that they are talking about.  The citation should include the author’s last names followed by the reference number. For example, (for reference [12]) the citation would be Yang, Zheng, Zhang and Zheng [12].  Once an article like this one with 3 or more authors has been cited, it can thereafter be cited as Yang et al. [12].  In the case of references (e.g., reference [15]) where there are 5 or more authors, the initial citation can be just the first author’s last name followed by et al. (e.g., Jia et al. [15]). Line 69: Did the authors in [12] use a BackPropogation NN that was already been trained for fire detection, or did they build one to train with this data? The tenses in the sentence are mixed, which makes this sentence unclear. Line 71: replace “Smoke detection is another attractive study, which…” with “Smoke detection is another means for fire detection, which…”. Line 72: replace “…improve the detection rate and detect fire in good manner.” With “…improve the detection rate and detect fire in reliable manner.” Line 77: “All above detection algorithms are based on…” should be replaced by “All above detection algorithms were based on…”. Line 78: The authors need a transition sentence for the types of methods you're talking about in this paragraph. The switch from conventional methods to deep learning methods is not clear in this paragraph. Line 82-83: “…synthetic smoke images to expand smoke dataset and domain adaption to train CNN structures” should be changed to “…synthetic smoke images to expand the smoke dataset and domain adaption to train the CNN structures” Line 85: “to expand dataset” change to “to expand the dataset” Line 86: “…it takes more time to make the data size meet the training…” should be changed to “…it takes an excessive amount of time to make the data size meet the training…” Line 88: it is not clear what the authors mean by “shorter connections”. Do they mean computationally shorter, shorter in terms of training iterations?  Please clarify mathematically or conceptually. Line 95: “…Cross Entropy Loss function, but it is easy to over-fitting when the 95 dataset is imbalance of class.” Replace with “…Cross Entropy Loss function, but it is easy to over-fit when there is dataset class imbalance.” Line 97: “…network using Focal Loss to learn the varying shapes of the lesions and demonstrated effective performance in lesion segmentation.” The English word “lesions” in this context seems incorrect so perhaps there was a translation issue? Lines 99-104: multiple errors in the paragraph. “Inspired by recent use of Focal Loss in image-based object detection, we improve the Cross Entropy Loss function to be suitable for wildfire smoke detection. Meanwhile, we added negative samples such as cloud, fog and haze as interference to our dataset. Unlike the aforementioned works, our network directly trains candidate smoke regions to eliminate the interference of complex background on feature extraction process. In addition, the problem of high false alarm rate and imbalance of training set can be solved simultaneously.” Change to the following, “Inspired by recent use of Focal Loss in image-based object detection, we improved the Cross Entropy Loss function to be suitable for wildfire smoke detection. We also added negative samples such as cloud, fog and haze as interference to our dataset. Unlike the aforementioned works, our network directly trained candidate smoke regions to eliminate the interference of complex backgrounds on the feature extraction process. In addition, the authors attempted to address the problems of high false alarm rates and imbalance of training sets simultaneously.” Lines 106-107: change “Our goal is to reduce the false alarm rate of detecting wildfire smoke and alleviate the imbalance of wildfire smoke dataset, and a novel framework integrating conventional method to CNN for the wildfire smoke detection is presented.” to the following, “Our goal was to reduce the false alarm rate of detecting wildfire smoke, alleviate the imbalance of wildfire smoke dataset, and undertake a novel framework integrating conventional method to CNN for the wildfire smoke detection.” Lines 108-110: The wording in the sentences is misleading. It wording makes it sounds like the smoke in the images is segmented or isolated before it is detected.  While it becomes clear in the sections that follow, the authors need to state that the segmentation is an initial pass at parsing the smoke from the background image and that the “detection” phase refines that classification. Line 112: change “result” to “results”. As well, is this from the research that you cite in the next sentence or from your own analysis? Line 113: change “[25-27] discussed smoke images in different color spaces.” to “Past research has discussed smoke images in different color spaces [25-27].” Line 115: change citation format for sentence beginning with “[23] has” to “Prema, Vinsley and Suresh [28] have…” Line 125: The authors state that the segmentation strategy eliminates the background. Assuming that column "E" in Figure 1 represents the image of the smoke segmented from the background, I still see a lot of background in some of those images. A qualification statement is needed to specify how much of the background is supposed to be eliminated by the segmentation process. Lines 129-132: replace “In order to further classification candidate smoke region, we proposed a novel deep learning architecture, i.e. dilated DenseNet, for wildfire smoke detection. We firstly introduced the major structure of DenseNet, then our proposed wildfire smoke dilated DenseNet(WSDD-Net) architecture is described in details.” with the following, “In order to further classify the candidate smoke regions, we proposed a novel deep learning architecture, i.e. dilated DenseNet, for wildfire smoke detection. Below we will introduce the major structure of DenseNet, followed by a detailed description of our proposed wildfire smoke dilated DenseNet(WSDD-Net) architecture.” Line 137: the authors need to define what a dense block is. Line 138: replace “…and the last one is connected to the global pooling…” with “…and the last layer is connected to the global pooling…” Line 147: replace “…multi-scale feature can be extracted…” with “multi-scale features can be extracted” Line 150: the formatting of the “3x3” has this text floating above the line Line 153: “… figure 4, Dilated…” Dilated shouldn’t be capitalized. Line 162: “architecture of existing framework” should be “architecture of the prior framework” (existing implies that the framework the authors used was incapable of multi-scale feature extraction.) Line 167-168: “It is designed for training imbalance wildfire smoke dataset by down-weighting of non-smoke dataset.” Should be changed to “It is designed for training to account for the imbalance in wildfire smoke datasets by down-weighting of non-smoke dataset.” Also, does down-weighting refer to the weights within the network? Line 169: Pt is never in the loss function…shouldn't y1 be replaced with Pt? Did you use a sigmoid function as your activation function? Line 170: So is alpha the ratio of smoke images/total images (where total images = smoke + not smoke)? You never define alpha. If alpha is less than one here, then you have a low penalty for false positives and false negatives…? Please clarify! Line 172: “…high detection accuracy rate and low false alarm rate.” Do you mean positives and false negatives here? Clarify Line 175: What does the 32 refer to in the phrase “mini-batch size of 32 on our two GPUs”? Is it 32 images or something else? Clarify Line 176: Please define what gamma is. Line 177: is the image size (224x224) sufficient? I see only 1 image where the smoke is far away but could it be influenced by the reduced image quality?  While I understand that authors didn't have many smoke images to use, some of the imagery seems to be from airborne sources and some from ground sources.  If the system were to be used for a given area to detect fires, what do the authors expect the image source to be (ground, satellite, airborne etc.)?  There are issues specific to each source that would need to be overcome by the model. Line 180: it would be helpful to indicate in the caption for Figure 5 what the images are. In the text that follows the figure, it is clarified that the top 2 rows are smoke images but it should also be put in the figure caption. Line 182: “and websites by ourselves.” Change to “and websites by the authors.” Line 182-183: “The 80% images are randomly separated according to the ratio of 3:1 for training and validation.” Should be changed to “80% of the images were randomly separated according to the ratio of 3:1 for training and validation.” Again, it is unclear whether this is 80% of the whole dataset or 80% of the smoke images and 80% of the non-smoke images. Lines 183-185: “The extra 20% images, which is prepared for the performance comparison between benchmarking algorithms and our framework, are adopted as testing set.” Should be changed to “remaining 20% of the images were prepared as testing set for the performance comparison between benchmarking algorithms and our framework.” Lines 185-186: “Rotation, Random-Crop and mirror were implementated to expand the size of training dataset.” Should be changed to “To expand the size of the training dataset, the images were rotated, randomly cropped and mirrored to generate additional images.” Were all three manipulations used on the images?  Also, were these manipulations used on all of the images, or only the fire images?  What was the final size of the image set used? Line 197: The term recall rate is a new one. The authors should indicate that it is defined in equation 8.  Line 201: Your training method was never defined…your datasets were (training vs validation), but never how the WSDD-net was trained. Also see line 183 <-- you mentioned a "validation" dataset, but not how it was used. Did you perform cross-validation during testing to check that the model isn't overfitting the data during training? Line 205-206: “Figure 6(a) shows the changing of training dataset accuracy with different epochs.” Should be “Fig. 6 shows the changing accuracy of the system with training over different epochs.” Line 207 & Line 226: This rise in learning is true of all of the learning networks, although the WSDD-Net appears to reach a higher accuracy than the other learning networks. It should also be noted that, other that the Inception-V3 network, the WSDD-Net only increases the accuracy by about 4-5%. Thinking about future use of the system and given the additional manipulations required for training with the WSDD-Net, is the additional 4-5% worthwhile?  Line 208: “…and fluctuates around 99%.” The authors should include a measure of variability such as a standard deviation. Line 211: “other classical network” should be “other classical networks” Line 212: “from statistical perspective”. The authors cannot claim this as they didn’t run any statistical analyses to prove that the WSDD-Net was better. Also, this is the training dataset, the actual worth of their network is in how it handled the testing dataset(s). Lines 215-216: “Furthermore, we noticed that almost all listed classical algorithms are not stable when dealing with the images of negative dataset…” The phrase "not stable" implies greater variation.  While the authors have presented the average FAR for their learning network and for the ResNet-101, they haven't provided any measures of variability (e.g., standard error or standard deviation).  Also, the authors need to clarify what they mean by a negative dataset. Are they referring to true negatives (in the signal detection theory sense)? Lines 219-220: “Since the DenseNey-121 and WSDD-Net have much smaller model size than other networks, and have the ability to adopt small dataset to prevent overfitting.” Should be changed to “The DenseNet-121 and WSDD-Net have much smaller model sizes than other networks, and have the ability to adopt small datasets to prevent overfitting.” Line 222: It is not completely clear if the authors are still discussing training data results or the results of the testing part of the dataset. Line 223: Again the F1 is only a little more than 2% better than the DenseNet-121. Does this constitute a significant improvement in the performance? Line 226: Did you do any gradient checking throughout the training process? Line 229: It is not clear what “Size” refers to in the table. Was it the size of the network or the size of the datasets? (I assume that it is the size of the learning network software but the authors should state that.) also, your FAR rate is far below that of the others. You need to explain why you found this. Line 231: The authors should use a different color (green, perhaps?) as I initially mistook the HLTPMC data for the WSDD-Net data in Figure 7. Line 242: “…wildfire smoke detection has better generalization capabilities…” Did the authors train the other networks on the WS dataset and then test them on the Yuan dataset? It wasn't mentioned.  If not, they cannot say that the WSDD-Net generalizes better since they did not test the generalization of the other networks. Line 255: “…cross entropy loss function is used…” change to “…cross entropy loss function was used…” The authors should also restate how their network was improved to differentiate it from the traditional loss function. Line 256: “The proposed WSDD-Net has been evaluated on two smoke datasets…” should be changed to “The proposed WSDD-Net was evaluated on the WS and Yuan smoke datasets…”

Author Response

On behalf of all the contributing authors, we would like to thank you for your constructive comments concerning our article (Manuscript ID: electronics-594066). These comments are all valuable and helpful for improving our article. All the authors have seriously discussed about all these comments. According to the comments, we have tried best to modify our manuscript within the document were all highlighted by using red coloured test. Point-by-point responses are listed below.

Reviewer 2 Report

Figure 1. The note of (f) needs to be added. Section 4. Please demonstrate what are the major situations of the commission error and omission error for your algorithm.  

Author Response

(The authors gave the same response as above.)

Round 2

Reviewer 1 Report

The authors have done a good job revising the manuscript.  The revised manuscript is easier to follow and understand.  There are still a few minor grammatical errors and 1-2 instances where the meanings of their sentences require clarification.

Unfortunately, there is still one major issue that they haven’t addressed sufficiently. It comes up in a few places in the manuscript where the authors indicate that their WSDD-Net results in better identification of smoke than other systems. It was mentioned in point 4 point 32, point 33, point 34, and point 36 of the author’s cover letter.  While the authors indicate that their model results in better performance for several measures, they haven’t included any statistical analyses of these improvements or any measures of the variability in their results.  While their model may actually prove to be better than other models, the way the manuscript is currently written suggests that any improvement over the existing models’ performance would be proof that their model is better.  The difficulty with this is that there is no guideline as to what constitutes an improvement, even if there are improvements on multiple measures.  If performance was improved by 1%, 0.1% or 0.001% one could argue that a model showing this improvement was better than other models. 

Ideally, the improvement could be shown by by running their algorithms on several different datasets (6-12 different datasets) to show that their model can outperform other models on a consistent basis.  Given that the authors cannot access that many datasets, this approach is impossible.  However, the authors should look at a paper by Ghaffari, Abdollahi, Khoshayand, Bozchalooi, Dadgar and Rafiee-Tehrani (2006). (Reference: A. Ghaffari, H. Abdollahi, M Khoshayand, I. Bozchalooi, A. Dadgar, & M. Rafiee-Tehrani (2006). Performance comparison of neural network training algorithms in modeling of bimodal drug delivery. International Journal of Pharmaceutics, vol. 327, pp.  126–138.) In this paper, the authors have presented a way to conduct a statistical performance comparison of different neural network training algorithms.  Ghaffari et al (2006) compared different training algorithms on the predictive ability of neural networks. To test which training method resulted in the best performance, they used the root mean square error of the neural network output classifications as a performance metric (much like your accuracy metric for WSDD-Net) and conducted a series of t-tests between each training algorithm, with 0.05 as a significance cut-off.

Using statistical methods like this one quantify the impact that one model has over another. For example, if a model is 15% better or 5% better than another model, we would want to know if those improvements are statistically significant so that we can say with confidence that the new model is better than others.

The t-test for independent samples is a fairly easy test to run and there should be many websites that the authors can access that describe the test and likely more than a few that will actually run the test and provide the results. The authors could also consult the math or psychology departments at their university to get input on the test if they need to get additional advice on running the test. If, after consulting your colleagues, they advise you to run a different statistical test, that would also be acceptable.

It took a while to track down a paper that actually did do a statistical comparison of different training algorithms.  My apologies to the authors that I was not able to find this for the first round of reviews.

Other minor points in the revised manuscript are as follows:

Line 42: “accuracy of many existing image-based smoke detection techniques is still vulnerable.” Change vulnerable to inconsistent.

Line 49: “…detection, pedestrian and vehicle images are not used as negative samples.” This still is not clear.  Are the authors talking about images taken by pedestrians & vehicles or images with pedestrians & vehicles in the image?  Also the double negative here (“not used as negative samples”) is hard to interpret.  Does this mean that they are used as positive samples?

Line 50: “colored regions are similar to smoke in color” change to “colored regions resulting from fog, clouds or haze are similar to smoke in color”

Line 51: change “detect smoke” to “discriminate smoke”

Line 53: delete the word “few”

Line 60: change “images using segmentation” to “images using a segmentation”

Line 70: change “based on Levenberg-Marquardt” to “based on the Levenberg-Marquardt”

Line 71:  change “In [13],…” to “In Zhang,  Shen, and Zou [13],…”

Line 78: “All above detection algorithms mainly used handcrafted and…” I think there is a word missing after handcrafted. I’m guessing that the word would be imagery?  

Line 79: “learning methods, while a cost-effective…” This should be split into 2 sentences. There should be a period after “methods”, the word “While” should be deleted and the word “a” should be capitalized to start the new sentence.

Line 80: “In [18],…” should be “In Mao, Wang, Dou, and Li [18],…”

Line 87: “the method of expanding” should be “this method of expanding”

Line 91: “more sufficient for training” should be “more efficient for training”

Line 92: “solve this problem and propose the DenseNet” should be “solved this problem and proposed the DenseNet”

Line 100: “identify more fine features” should be “identify finer features”

Line 101: “by focusing on hard classify examples”  I assume the authors mean “hard-to-classify examples”?

Line 111: “CNN for the wildfire smoke detection” should be changed to “CNN for wildfire smoke detection”

Lines 112-113: “pre-processing method of smoke” should be changed to “pre-processing method of separating smoke”

Line 128: “removes the complex background except for smoke colored regions.” change to “removes the complex background except for smoke colored regions which may include elements like clouds, haze or fog.”

Line 140: “Figure 2 replace the common convolutions in original DenseNet” change to “Figure 2 replaces the common convolutions in the original DenseNet”

Line 162: “As shown in figure 4. dilated” change to “As shown in Figure 4, dilated”

Line 177: “down-weighting of non-smoke dataset” change to “down-weighting the non-smoke part of the dataset”

Line 188: “encompasses an E5-2620 CPU” change to “consists of an E5-2620 CPU”

Line 201: “Rotated and mirrored were used” change to “Rotation and mirroring were used”

Line 217: “Recall Rate (RR) were” change to “Recall Rate (RR, defined below) were”.  Again the other variables (TP, TN, FP, & FN) are standard to signal detection.  The authors need to let the reader know that it will be defined in the equations below this paragraph.

Line 223: “Recall Rate (RR) were involved” change to “Recall Rate (RR) were also used”

Line 242: “DenseNey-121” should be change to “DenseNet-121”

Line 275: “an improvement cross entropy loss function” should be “an improved cross entropy loss function”

Lines 275-276: “an improvement cross entropy loss function was used instead of cross entropy loss function in the training process.”  Do the authors mean that an improved cross entropy loss function was used instead of a regular cross entropy loss function?  Even if that is the case, they need to add a word or two to describe how the cross entropy loss function was improved.

Author Response

Thank you for your letter and constructive comments concerning on our article (Manuscript ID: electronics-594066). These comments are all valuable and helpful for improving our article. All the authors have seriously discussed about all these comments. According to the comments, we have tried best to modify our manuscript within the document were all highlighted in yellow. Point-by-point responses are listed below.

Round 3

Reviewer 1 Report

The authors have again done an excellent job incorporating all of the changes in the manuscript there are only a few minor points that need to be addressed.

Line 50: "images with pedestrian and vehicle are not used" should be changed to "images with pedestrians and vehicles are not used"

Lines 53-54: "There are wildfire smoke datasets because there are not many few wildfires and so smoke images cannot..." should be changed to "There are not many wildfires and so wildfire smoke images cannot..."

Line 57: "we propose a novel framework" should be changed to "we proposed a novel framework"

Line 80: "All above detection algorithms..." should be changed to "All of the above detection algorithms..."

Line 90: "...expanding dataset with synthetic..." should be changed to "...expanding datasets with synthetic..."

Lines 259-260: The authors need to specify what dependent variable was used for the analyses. I assume since they referenced Ghaffari that they looked at the RMS for their model compared to the other models They can also add a sentence to talk about their results which can indicate that the WSDD-Net had lower RMS error that the other models based on the statistical analysis. 

Lines 267-268:  The notation in the final column of Table 2 is a little confusing. I assume that the authors meant that the null hypothesis was true or false. I would suggest that they remove the final true/false column in the table and put rows 3-7 in bold.

Author Response

(The authors gave the same response as above.)
